# New Electrospun ZnO:MoO_3_ Nanostructures: Preparation, Characterization and Photocatalytic Performance

**DOI:** 10.3390/nano10081476

**Published:** 2020-07-28

**Authors:** Petronela Pascariu, Mihaela Homocianu, Niculae Olaru, Anton Airinei, Octavian Ionescu

**Affiliations:** 1“Petru Poni” Institute of Macromolecular Chemistry, 41A Grigore Ghica Voda Alley, 700487 Iasi, Romania; mihaela.homocianu@icmpp.ro (M.H.); nicolae.olaru@icmpp.ro (N.O.); anton.airinei@icmpp.ro (A.A.); 2National Institute for Research and Development in Microtechnologies-IMT Bucharest, 126A, Erou Iancu 8 Nicolae Street, 077190 Bucharest, Romania

**Keywords:** molybdenum trioxide-incorporated ZnO, structural characterization, optical properties, photocatalytic activity

## Abstract

New molybdenum trioxide-incorporated ZnO materials were prepared through the electrospinning method and then calcination at 500 °C, for 2 h. The obtained electrospun ZnO:MoO_3_ hybrid materials were characterized by X-ray diffraction, scanning and transmission electron microscopies, ultraviolet (UV)-diffuse reflectance, UV–visible (UV–vis) absorption, and photoluminescence techniques. It was observed that the presence of MoO_3_ as loading material in pure ZnO matrix induces a small blue shift in the absorption band maxima (from 382 to 371 nm) and the emission peaks are shifted to shorter wavelengths, as compared to pure ZnO. Also, a slight decrease in the optical band gap energy of ZnO:MoO_3_ was registered after MoO3 incorporation. The photocatalytic performance of pure ZnO and ZnO:MoO_3_ was assessed in the degradation of rhodamine B (RhB) dye with an initial concentration of 5 mg/L, under visible light irradiation. A doubling of the degradation efficiency of the ZnO:MoO_3_ sample (3.26% of the atomic molar ratio of Mo/Zn) as compared to pure ZnO was obtained. The values of the reaction rate constants were found to be 0.0480 h^−1^ for ZnO, and 0.1072 h^−1^ for ZnO:MoO_3_, respectively.

## 1. Introduction

Currently, many efforts are being made worldwide to develop new high-performance photocatalytic materials for energy and environmental applications. Metal oxide semiconductor materials in various shapes and structures, including ZnO, TiO_2_, CuO, and MgO, have proven to be a good alternative for the degradation of various organic dyes. It is known that ZnO is an oxide semiconductor having a broad direct band gap (3.37 eV), high excitation binding energy (60 meV), and good electrical, mechanical, and optical and photocatalytic properties, comparable to those of TiO_2_. Metal doping of ZnO can significantly improve the photocatalytic activity, it is thought by (i) generation of trapping site which will decrease the recombination rate of photoinduced electron-hole pairs; (ii) decrease of band gap energy of photocatalysts; and (iii) structural defects in the crystalline phase of ZnO. Many studies have been based on the development of new photocatalysts based on ZnO doped with different metals (Ag, La, Er, Sm, Cu, Au, Ce, Ni, Fe, etc.) to improve the photocatalytic activity for the degradation of different organic dyes and to extend the degradation domain using visible light [1,2]. For example, Pascariu et al. reported good photocatalytic responses of ZnO-SnO_2_ nanostructures used for rhodamine B dye degradation with an initial dye concentration of 0.01 mM and a catalyst dosage of 0.5 g/L [3]. Likewise, the same authors obtained an improvement of the photocatalytic activity after doping ZnO with Ni or Co [4]. Molybdenum trioxide (MoO_3_) is a very interesting transition metal oxide having a wide band gap energy of about 3 eV, distinctive optical properties, and highly visible-light photocatalytic activity [5,6]. But the synthesis of ZnO-MoO_3_ nanostructures (especially by the electrospinning method) and their detailed properties after being incorporated into the ZnO matrix are less reported compared to other semiconductor oxides, such as SnO_2_, ZnO, TiO_2_ and In_2_O_3_ [7,8]. Moreover, it is known that the properties of molybdenum trioxide are strongly dependent on the synthesis methods and can be greatly modulated by doping it with other metal oxides [9]. There are many methods to develop new materials of different shapes and sizes, including hydrothermal, sol-gel, precipitation, microemulsion, solvothermal, the electrochemical deposition process, microwave, polyol, wet chemical method, flux methods and electrospinning [10]. The electrospinning method is one of the simplest, cheapest, and most efficient for obtaining nanostructured materials. Also, this method offers the possibility of obtaining materials with controllable diameters of the fibers, very high surface-to-volume ratio, and specific surface and excellent functional properties [1]. Also, the electrospinning technique is intensively used in obtaining ceramic materials. One study based on Mo-doped ZnO materials obtained by the electrospinning method has been reported by Kim et al. [11], and their investigation regarding gas-sensing properties in ethanol, trimethylamine (TMA), CO and H_2_ medium. To the best of our knowledge, the development of ZnO:MoO_3_ nanostructures by the electrospinning-calcination method and then their testing for dye degradation have not been reported in the literature so far. Similar systems have been obtained by other methods, such as coating of MoO_3_ altered ZnO, by surface metal impregnation [12], ZnO@MoO_3_ core/shell nanocables by the electrodeposited method [13], and 1D/1D ZnO@h-MoO_3_ synthesized via the solid state impregnation-calcination method [14].

Therefore, in this paper, we proposed the preparation of new pure ZnO and ZnO:MoO_3_ nanostructures by the electrospinning method and then calcination at 500 °C for 2 h. The polymer solutions were prepared from polyvinyl alcohol (PVA) dissolved in water. The details of the structural and optical properties of the obtained MoO_3_ incorporated-ZnO nanostructures were comparatively analyzed and discussed. Moreover, the photocatalytic performance of these materials was evaluated by photodegradation of RhB dye in aqueous solution under simulated sunlight irradiation.

## 2. Materials and Methods

### 2.1. Materials

Polyvinyl alcohol (PVA) (Mn = 80.000), zinc acetate [Zn(CH_3_COO)_2_·2H_2_O], and ammonium molybdate ((NH_4_)_6_Mo_7_O_24_·4H_2_O) were purchased from Sigma-Aldrich (Merck KGaA, Darmstadt, Germany). Deionized water was used as a solvent.

### 2.2. Preparation of ZnO:MoO_3_ Nanostructures

We dissolved 0.55 g PVA powder in 5 mL of deionized water and then heated at 100 °C under vigorous magnetic stirring for 8 h and thus the solution for electrospinning was obtained. This solution was cooled to room temperature and then 0.4 g of zinc acetate was added to it. Finally, ammonium molybdate in various concentrations was added to the above electrospun solution. This prepared viscous solution was stirred for another 3 h and then transferred into a needle syringe with a diameter of 0.5 mm. All new nanostructures were obtained using a home-made electrospinning device [15,16]. The optimal conditions for the production of these fibers before calcination were as follows: high voltage source (25 kV), the distance between the needle tip and the collector (stainless steel foil) was 15 cm at a flow-rate of about 0.75 mL/h. The ZnO:MoO_3_ nanostructures were obtained after calcination in an oven (in air) at 500 °C for 2 h. In this study, a series of four materials (with different concentrations of MoO_3_) were prepared and are presented. More details regarding growth and sample names correlation with the specific growth conditions can be found in Table 1.

### 2.3. Structural Characterization

X-ray diffraction (XRD) patterns of ZnO:MoO_3_ nanostructures were recorded using a Rigaku SmartLab-9kW X-ray diffractometer (Rigaku Corporation, Japan). SEM (scanning electron microscopy)/energy-dispersive X-ray spectroscopy (EDX) measurements were performed using a JEOL JSM 6362LV (Japan) electron microscope coupled with an EDAX INCA X-act Oxford Instrument detector (Oxford, UK). Transmission electron microscopy (TEM) studies were performed using a Hitachi HT7700 Dual Mode STEM (Japan) in TEM mode. To perform the TEM studies, a small quantity of material was fixed onto Cu mesh grids after dispersion in ethanol and sonication.

### 2.4. Steady-State Spectral Measurements

The optical measurements of the samples were evaluated by ultraviolet–visible (UV–vis) reflectance spectra recorded with a SPECORD 210Plus Analytik Jena (Jena, Germany) spectrophotometer equipped with an integrating sphere. The band gap energy values were found from the optical data solution. Steady state absorption and fluorescence spectra were recorded with a spectrophotometer SPECORD 210Plus Analytik Jena (Jena, Germany) and on an Edinburgh FLS980 spectrometer (Edinburgh, UK), respectively.

### 2.5. Time-Resolved Fluorescence

The time-resolved fluorescence spectra were collected on an Edinburgh FLS980 spectrometer using a time correlated single photon counting method. The emission decay profiles for all nanostructures dispersed in 1-propanol solution were determined in a 10 × 10 mm quartz cell, excited by a nanosecond diode laser (EPL-375) (Edinburgh Instruments Ltd., Livingston, UK) operating at 375 nm as light source. The fluorescence decays were evaluated using the nF9000 software attached to the equipment (Edinburgh Instruments Ltd., Livingston, UK) and the best fitted parameters were obtained for the reduced chi-squared values close to 1, and the weighted residuals were uniformly distributed around the zero line.

### 2.6. Nanosecond Transient Absorption Spectroscopy

Nanosecond transient absorption experiments were performed using a nanosecond laser flash photolysis technique (LP980, Edinburgh Instruments, (Livingston, UK)). The spectrometer was connected to a laser source (Ekspla NT342), which allows us to generate a high concentration of excited state species, laser pulses at 372 nm, and a frequency of 1 Hz. The absorption lifetime values of excited state species were evaluated with the L900 software attached to the equipment ((Edinburgh Instruments Ltd., Livingston, UK).

### 2.7. Photocatalysis Tests

The photocatalytic activity of the MoO_3_-incorporated ZnO nanostructures was tested by photodegradation of rhodamine B (RhB) dye in aqueous solution under visible light irradiation using the same experimental degradation procedure as reported in previous work [3]. Briefly, 5 mg of the catalyst was dispersed in a vial containing 10 mL of RhB dye solution (5 mg/L), having a controlled temperature at 25 °C. A 100 W tungsten lamp was served as the light source. The power of light source was 102.74 kJ·m^−2^·h^−1^, and measured by a PMA 2100 apparatus, prod by Solar Light Co. (Glenside, PA, USA). The wavelength range of the tungsten bulb varies between 350 and 850 nm. The initial concentrations of RhB dye and after irradiation at different times were determined using a UV–vis spectrometer SPECORD 210Plus, Analytik Jena (Jena, Germany).

## 3. Results and Discussion

### 3.1. Morphological Characterization

Some examples of SEM micrographs of ZnO:MoO_3_ materials obtained after calcination at 500 °C for 2 h are shown in Figure 1. Figure 1a,b depict the morphology of pure ZnO and MoO_3_ metal oxides. The as-grown ZnO exhibits a cylindrical microrods structure with a rough surface. The SEM image (Figure 1b) corresponding to the MoO_3_ sample reveals a morphology composed of microparticles, with a platelet structure and diameters between 4–5 µm. This different structure of the two types of materials (ZnO and MoO_3_) leads to a morphology composed of different types of crystals interconnected between them in the ZnO:MoO_3_ nanostructures. TEM studies were performed to confirm the intimate structure of the ZnO:MoO_3_ materials. TEM observation proved the formation of two kinds of well-shaped nanosized crystallites: one with dimensions of about 20–30 nm and the second, with larger size of ~40–45 nm. An example of TEM micrographs of nanocrystalline building blocks on sample S5 is presented in Figure 2.

EDX analysis was used to report the elemental composition of all samples. The EDX spectra (Figure 3) confirm the presence of Zn and O in a ratio of about 1:1, corresponding to the ZnO sample and the presence of Mo and O in a ratio of 1:3 corresponding to the MoO_3_ sample. The atomic percentages of Mo in composite nanostructures were: 0.92% (S3), 1.51% (S4), 2.30% (S5), and 4.82% (S6), respectively (Table 1). Also, the EDX measurements have predicted the expected values for Mo according to the prepared samples.

### 3.2. X-ray Diffraction (XRD) Analysis

The crystalline structure of pure ZnO, MoO_3_ and ZnO:MoO_3_ composites were confirmed using XRD measurements, and the XRD patterns are illustrated in Figure 4. The diffraction peaks of ZnO are indexed as hexagonal wurtzite-type structure (JCPDS No. 00-230-0112, space group: P63mc, No. 186) having as main peaks of diffractions at 2θ = 31.84°, 34.51°, 36.32°, 47.62°, 56.66°, 62.92°, 66.43°, 67.98° and 69.13°, attributed to diffraction planes (110), (002), (101), (102), (110), (103), (200), (112), (201), respectively. The diffraction pattern of α-MoO_3_ confirms the orthorhombic crystalline structure Joint Committee on Powder Diffraction Standards (JCPDS No. 00-900-9670, space group: Pbnm, No. 62) with the main diffraction peaks at 23.36°, 25.65°, 27.23°, 29.59°, indexed by (110), (040), (021) and (130). The XRD data corresponding to the ZnO:MoO_3_ composites show a mixed crystalline structure formed from ZnO with hexagonal wurtzite structure and MoO_3_ with orthorhombic structure. The average crystallite size was calculated according to the Scherrer relation *D* = 0.8*λ*/*βcos*θ, were λ is the X-ray wavelength corresponding to CuKα radiation, β is the full width at half maximum of the peak and θ is the Bragg angle. The average crystallite size was found to be D = 21.77 nm for pure ZnO and D = 14.93 nm is corresponding to ZnO:MoO_3_ nanostructures, respectively. The presence of reflections corresponding to ZnO and MoO_3_ in ZnO-MoO_3_ nanostructures shows the successful formation of the nanocomposite. This decrease in the crystallite size of ZnO:MoO_3_ (S5) may be attributed to the formation of Mo–O–Zn bands on the surface of the doped materials, which inhibits the growth of the crystallite, as reported by many authors for similar systems [17,18]. The decrease in the crystallite size of ZnO:MoO_3_ nanostructures can be determined by the lattice distortion by Mo incorporation due to the difference between the ionic radius of Mo^6+^ (0.065 nm) smaller them that of Zn^2+^ (0.074 nm). The lattice constants *a* and *b* were 3.252 Å, 5.209 Å for ZnO and 3.249 Å, 5.207 Å obtained for ZnO:MoO_3_ (S5), values that are in good agreement with the standard ones (a = 3.253 Å, c = 5.213 Å for ZnO → JCPDS 34-1451) [19].

### 3.3. Optical Properties

Evaluation of optical properties of materials is a very important factor for the study of their photocatalytic activity. UV–vis absorption and emission spectra of MoO_3_-incorporated ZnO nanostructures dispersed in 1-propanol solution were investigated at room temperature. The incorporation of MoO_3_ into the ZnO matrix modify the optical properties of the obtained nanostructures. Figure 5. showed the absorption spectra of the pure ZnO- and MoO_3_-incorporated ZnO samples with various MoO_3_ weight percentages (S1 → S5). It can be observed that the absorption of all the MoO_3_-incorporated ZnO samples did not show a large difference in the UV region (only small variations (1–2 nm) in the absorption band position in the spectra). In contrast, the absorption maxima of all MoO_3_-incorporated ZnO samples shift towards blue (from 382 to 371 nm) are observed, compared with the absorption maximum of the pure ZnO samples. These shifts are due to the incorporation of the MoO_3_ materials in the pure ZnO matrix. From Figure 5, it can be remarked that the absorption of ZnO:MoO_3_ nanocomposites was enhanced when the MoO_3_ doping level did not exceed 1.66% molar ratio Mo/Zn, while for nanostructures with a higher content of MoO_3_ the absorption shows a small decrease (Figure 5).

One of the important effects of the MoO_3_ oxide incorporation into the ZnO matrix is the reduction of the direct band gap energy value from 3.211 eV (corresponding to ZnO) to 3.170 eV for ZnO:MoO_3_ composite materials (Table 2). The band gap energy values of the pure ZnO, MoO_3,_ and for the ZnO:MoO_3_ composite materials were determined by UV–vis measurements using reflectance spectra, registered between 300–600 nm. Figure 6a showed the reflectance spectra corresponding to pure materials (ZnO, MoO_3_) and composite systems ZnO:MoO_3_ (S3 → S6). Thus, the optical band gap energy values, E_g_, for all materials were calculated using the Kubelka–Munk equation presented below:(1)$$F\left(R\right)=\frac{{\left(1-R\right)}^{2}}{2R}$$
where *F*(*R*) is the Kubelka–Munk function and *R* is the diffuse reflectance of materials, respectively. Additionally, using $\left[F\left(R\right)h\vartheta \right]=c{\left(h\vartheta -E_{g}\right)}^{n}$ equation: where hϑ is photon energy, *n* is a constant giving the type of optical transition, *c* is a constant and *E_g_* denotes the band gap energy.

The plots of ${\left[F\left(R\right)h\vartheta \right]}^{2}$ as a function of the photon energy (hϑ) of all the materials are presented in Figure 6b and by extrapolating the linear part of each curve, the values of the direct optical band gap (Eg) were obtained. The pure ZnO material has an Eg value of 3.21 eV, this value is in good agreement with the reported literature for ZnO [20]. Instead, for pure MoO_3_ material, a lower Eg value of 2.96 eV (closer to the literature data) [9] was obtained. The band gap energies of the composite materials based on ZnO:MoO_3_ were calculated to be: 3.186 eV (S3), 3.177 eV (S4), 3.170 eV (S5), and 3.189 eV (S6), respectively. The presence of MoO_3_ dopant in the ZnO matrix induces a small shift of the absorption edge of ZnO (see Figure 6) in the long wavelength direction and thus decreases the band gap energy (Eg) values. Based on the values of Eg one can calculate the positions of the valence band (VB) and the conduction band (CB) (redox abilities) of new ZnO:MoO_3_ nanostructures, which directly influence their photocatalytic activity. The conduction band (CB) and valence band (VB) potentials of ZnO:MoO_3_ were calculated using the following equations [21].
E_VB_ = χ − E_e_ + 0.5E_g_(2)
E_CB_ = E_VB_ − E_g_(3)
where χ and E_e_, are the electronegativity (for ZnO, the value is approximately 5.79 eV) [22] and the free electron energy on the hydrogen scale (approximately 4.5 eV), respectively. The calculated values of E_VB_ and E_CB_ are listed in Table 2. The values of E_CB_ for MoO_3_-incorporated ZnO nanostructures shifted to lower energy compared to the pure ZnO, which caused a narrower band gap.

To explore the charge carrier trapping, migration, and charge transfer transitions in the ZnO:MoO_3_ hybrid nanostructures, the emission spectra of these new materials were recorded under 375 nm wavelength excitation (see Figure 7).

The emission spectra of MoO_3_-incorporated ZnO nanostructures show a sharp and intense blue emission band around 412–416 nm, ascribed to the near band edge (NBE) emission due to free exciton recombination [23] and a broader and lower green emission band in the visible region at ≈ 450–550 nm (with maxima around 506 nm). This green emission band is due to the formation of oxygen vacancies by the presence of MoO_3_ oxide in the structure of the target nanostructures and the presence of some intrinsic defects in the ZnO structure. Generally, the intrinsic defects in ZnO include oxygen vacancies (V_o_), zinc vacancies (V_Zn_) oxygen (O_i_), zinc (Zn_i_) interstitials, and defect states dominating the emission in the visible range [24]. It was observed from Figure 7 that the emission intensity is considerably enhanced with the addition of MoO_3_ as compared to the free ZnO sample. This remarkable improvement in the emission of MoO_3_ incorporated samples can be attributed to the energy transfer from ZnO to the MoO_3_ metal oxide and a higher recombination rate in these samples. Instead, in the photoluminescence spectrum of pure ZnO nanostructures, the wavelength of the maxima of the emission band is located at 431 nm, with low intensity, and the green emission band has disappeared (compared to the spectra of MoO_3_-incorporated ZnO samples). The undoped zinc oxide nanostructure displays only an emission band around 431 nm, with low intensity and the green emission band practically disappeared in comparison with MoO_3_-incorporated ZnO samples. Also, it can be seen that the intensity of the green emission band increases as the molybdenum trioxide level increases, excepting nanostructure ZnO:MoO_3_ (S5), where an emission band at about 478 nm has occurred (Figure 7), which have not been observed in the other samples due to the intense transitions determined by the oxygen vacancies. Moreover, the lower intensities of the green emission band in sample S5 can determine the creation of more electron trapping sites, which facilitates the transfer of photogenerated charge carriers increasing thus the electron-hole lifetime and the photocatalytic response was exchanged [25]. As seen from Table 3, the nanostructure ZnO:MoO_3_ (S5) has a larger decay time as compared to other samples. In fact, for this nanostructure the highest shift to shorter wavelengths of the absorption and emission bands was observed and at the same time, the lowest value of E_g_ was obtained (Table 2). The weak blue emission band at 478 nm can be assigned to the electron transitions between interstitial zinc (Zn_i_) and Zn vacancies [26]. Thus, the sample ZnO:MoO_3_ (S5) practically exhibits a very weak green emission band because the MoO_3_ loading can increase distortion centers and some surface defects in the lattice leading to the decrease of the oxygen vacancies. Furthermore, the emission spectra of zinc oxide (ZnO) sample incorporating with molybdenum trioxide (MoO_3_) appear similar in shape to the emission peaks but show a small shift (19 nm) in wavelength toward the blue range as compared to that observed for pure ZnO nanostructures.

### 3.4. Time-Resolved Photoluminescence Data and International Commission on Illumination (CIE) Coordinates

For this purpose, the samples (S1 → S6) were exciting with a light source of 375 nm. Base on the maximum emission band at 406 nm, the lifetime decay curves were evaluated. As a representative example, in Figure 8 is displayed the lifetime decay profiles for the ZnO:MoO_3_ (S5) sample. The decay curves were fitted to a bi-exponential equation for all the studied ZnO:MoO_3_ nanostructures and the experimental emission lifetime values of these materials were calculated using the equations given below [23]:(4)$$I\left(t\right)=A_{1}\exp\left(-\frac{t}{\tau_{1}}\right)+A_{2}exp\left(-\frac{t}{\tau_{2}}\right)$$
and the value of average decay time can be estimated by the following formula:(5)$$\tau_{eff}^{*}\;\left(ns\right)=(A_{1}\tau_{1}^{2}+A_{2}\tau_{2}^{2})/\left(A_{1}\tau_{1}+A_{2}\tau_{2}\right)$$
where *I*(*t*) is the time dependent emission intensity (emission intensity at any time); *A*_1_ and *A*_2_ are the fitting constants and have values between [0, 1]; *τ*_1_ and *τ*_2_ are the decay times of the fast and slow decay components.

The obtained values of the fast and slow decay components, as well as average lifetime values for all samples, are listed in Table 3. In general, the fast decay component is related to the non-radiative recombination and the slow decay is connected with the radiative lifetime of the free exciton. It was stated that the non-radiative recombination process is featured in defects related to oxygen vacancies [27]. The emission decay for all investigated nanostructures was obtained by using a biexponential model to fit the experimental data and have resulted in two lifetimes τ_1_ and τ_2_. For all investigated nanostructures τ_1_ is lower than τ_2_, but the contributions of the two lifetime components (a_1_ and a_2_) are opposite (Table 3). Therefore, the highest lifetime τ_1_ was obtained for the ZnO:MoO_3_ (S5) sample (0.465 ns). Both fast and slow decay constants can be tuned by adding MoO_3_ to the ZnO matrix. The irregular variations of lifetime values as a function of the MoO_3_ content can be explained by the difference of dispersion of MoO_3_ in these samples, where the redundant MoO_3_ can act as recombination centers [28]. Since MoO_3_ has a lower band gap compared to ZnO will induce a shorter lifetime for the generated electron-hole pairs, decreasing the efficiency. For the pure ZnO sample, the average lifetime values are shorter than that of all ZnO:MoO_3_ nanostructures (Table 3). This increase in the average lifetime values may be attributed to the MoO_3_ doping-induced non-radiative recombination centers. Moreover, the average lifetime values (Table 3) decrease with increasing doping concentration (from S1 to S5), which indicates the existence of an energy transfer process between the two components.

Based on the emission spectra of the pure ZnO and MoO_3_-incorporated ZnO nanostructures was built their CIE (International Commission on Illumination)-1931 color chromaticity diagram, which is shown in Figure 9. The values obtained for the color CIE chromaticity coordinates for pure ZnO and all the hybrid composites are shown in Figure 9. Generally, the quality of any light emitted can be investigated in terms of correlated color temperature (CCT) which can be calculated using the McCamy’s relation as [29]:CCT = 437n^3^ + 3601n^2^ + 6861n + 5517(6)
where, n = (x − xe)/(ye − y) is the inverse slope line and the point at xe = 0.332, ye = 0.186 is the epicenter. The values obtained for calculated CCT were found to be: 82,868 K, 11,389 K, 187,053 K, and 16,098 K, for S3, S5, S5, and S6 nanostructures, respectively. These values are located inside of the range of clear blue poleward sky light [30]. However, based on the chromaticity diagram, it can be said that the samples exhibit some differences in color emission determined by the MoO_3_ loading. As seen, the nanostructures containing 0.84% and 3.26% molar ratio Mo/Zn presented emissions in the blue region, whereas the samples having 1.66% and 4.81% molar ratio Mo/Zn exhibited coordinates close greenish-blue color. This color shift to the green region can be connected to a higher level of defects due to the high intensity of the green emission band around 520 nm, leading to a higher electron-hole pair recombination rate, which is not favorable to the photocatalysis process.

### 3.5. Nanosecond Time-Resolved Absorption Data

For two selected samples, (S3 and S5), the transient absorption data in 1-propanol solution were recorded at room temperature in order to monitor the optical absorption of photogenerated transient species (such as excited states, radicals and solvated electrons produced by the interaction between radiation and the substrate, more specifically, the kinetics of each transient species). Figure 10 shows the kinetics data, for S3 (Figure 10a,b) and S5 (Figure 10c,d) samples, which were fitted to exponential decay using Edinburgh Instruments software, after recording the decay rates of the transient species monitored at 372 nm (the wavelength at which these transient species absorb, see Figure 5). The decay rate for the S5 sample was found to be faster than the decay rate obtained for the sample having S3 (107.44 ns).

### 3.6. Photocatalytic Properties

The photocatalytic activity of the pure ZnO and synthesized ZnO:MoO_3_ nanocomposites was evaluated by photodegradation of rhodamine B (RhB) dye in aqueous solution, with an initial concentration of 5 mg/L, under visible light irradiation from a tungsten lamp (100 W)), at different times. The absorption spectra of the RhB dye (blank test) were analyzed at different exposure times (up to 20 h of irradiation), in the absence of the photocatalyst. The blank test showed that the intensity of the absorption bands of RhB decreases slightly reaching a degradation of about 5.8%. To get the most reliable results, the RhB dye degradation processes were analyzed in the presence of pure ZnO and compared with the ZnO:MoO_3_ (S3 → S6) hybrid materials. Figure 11 shows changes in the absorption spectra of RhB dye during irradiation under visible light in the presence of ZnO (Figure 11a) and ZnO:MoO_3_ (S5) (Figure 11b) nanostructures.

The intensity of the absorption bands in the absorption spectra of the RhB dye solution shown in Figure 11a,b decreases with increasing illumination time. It can be observed that the intensity of the absorption band at 556 nm decreased with increasing irradiation time, but with a maximum decrease observed for the S5 sample (RhB dye was highly degraded in this case). The removal efficiency (RE(%)) was calculated based on the following equation:(7)$$RE\left(\%\right)=\frac{\left(C_{0}-C_{e}\right)}{C_{0}}\times 100\%$$
(where *C*_0_ is the initial RhB concentration (mg/L) and *C_e_* is the RhB concentration at the time t (mg/L)) and the results obtained after the degradation of RhB dye for 6 h of irradiation in the presence of pure ZnO and ZnO:MoO_3_ are presented in Figure 11c. The removal efficiency of all samples obeys the followed order of S5 > S4 > S3 > S6 > pure ZnO (S1), that was very close to fluorescence data (fluorescence intensity). The higher photocatalytic efficiency of the ZnO:MoO_3_ (S5) nanostructures can be accounted for by decreasing Eg values compared to pure ZnO, a fact confirmed by other authors in differently doped ZnO-based systems [31,32]. Moreover, the increase in photocatalytic activity with the content of MoO_3_ may be due to the formation of MoO_3_/ZnO heterojunctions, as reported by Hirotaka et al. [33] for a similar system (TiO_2_/MoO_3_).

To estimate quantitatively the kinetics of RhB dye degradation, we used a pseudo-first order model expressed by the following equation: ln(C_0_/C) = kt, (where, C_0_ and C are the concentrations of dye in the solution at time 0 and t, respectively, and k is the pseudo-first-order rate constant). From the linear plots of ln(C/C_0_) versus the irradiation time (see Figure 11d), having a good correlation with the pseudo-first order reaction kinetics (R^2^ > 0.99), the values of the reaction rate constants were calculated and were found to be: k_1_ = 0.0480 h^−1^ (ZnO) and k_2_ = 0.1072 h^−1^ (ZnO:MoO_3_ (S5)), respectively. Likewise, these results show that after doping with 3.26% molar ratio Mo/Zn, the value of the reaction rate constant increases significantly, doubling its value (k_2_ = 2 × k_1_), as compared to that obtained for pure ZnO. The efficiency of photocatalytic degradation is found to be better in the presence of the MoO_3_-incorporated ZnO nanostructures than those of pure ZnO and/or MoO_3_ nanostructures [17], due to the reduction in band gap energy. The presence of structural defects due to the oxygen vacancies (V_O_), Zn-vacancy (V_Zn_), oxygen interstitial (O_i_), Zn-interstitial (Zn_i_) and the extrinsic impurities, which have been confirmed by photoluminescence measurements, also leads to an increase of photocatalytic efficiency for ZnO:MoO_3_ (S5) sample. Similar behavior has been also reported by other authors for the Ce, Gd doped ZnO nanostructures [34,35,36]. In addition, the photocatalytic efficiency may also be due to the microstructural defects that arise in the nanostructures after doping/loading with other materials. Table 4 presents the main works on Mo-doped ZnO materials present in the literature regarding photocatalytic activity for the degradation of different dyes. These are compared to the materials (ZnO:MoO_3_) presented in this work. According to the data listed in Table 4 [14,27,36,37], ZnO:MoO_3_ (with 3.26% molar ratio Mo/Zn into ZnO matrix) showed a good value of the reaction rate constants (k = 0.1072 h^−1^ or 0.0018 min^−1^) related to other studies for similar materials. However, the advantages of these materials are degradation of dyes in visible light of low intensity, the use of a reasonable amount of catalyst (0.5 g/L), and testing the material in soft conditions without acidification of solutions, in the absence of H_2_O_2_, and degradation of dyes at room temperature, usually used to boost the photochemical reactions.

## 4. Conclusions

Pure ZnO, MoO_3_ and MoO_3_-incorporated ZnO nanostructures were prepared by electrospinning and calcination at 500 °C for 2 h. The XRD diffractograms confirmed the hexagonal wurtzite-type structure for pure ZnO, the orthorhombic crystalline structure obtained for MoO_3_, and a combination of these structures for the ZnO:MoO_3_ (S5) composite nanostructure. The optical properties of the prepared MoO_3_-incorporated ZnO samples were studied using UV–vis absorption spectroscopy, and steady-state/time-resolved fluorescence spectroscopy. The absorption and emission bands are blue-shifted as MoO_3_ concentrations increase. The fluorescence lifetime decay analysis exhibits a bi-exponential equation, indicating the existence of the energy transfer processes, and the values of the average emission lifetime decrease with increasing MoO_3_ concentration. Moreover, the excited-state dynamics of S3 and S5 were characterized by nanosecond transient absorption spectroscopy. The photocatalytic activity of the pure/MoO_3_-incorporated ZnO nanostructures obtained was tested for the use in photocatalytic degradation of the aqueous rhodamine B dye solution under visible light irradiation. The values of the reaction rate constant obtained for ZnO:MoO_3_ sample (S5) was higher as compared to that of pure ZnO (S1) and depends on the MoO_3_ concentration (upon the incorporation of 3.26% molar ratio Mo/Zn into ZnO matrix, the value of reaction rate constant doubled). The present contribution is the first report on synthesis of these materials by electrospinning followed by calcination as well as study of photocatalytic activity under visible light for rhodamine B dye degradation.

## Figures and Tables

**Figure 1 nanomaterials-10-01476-f001:**
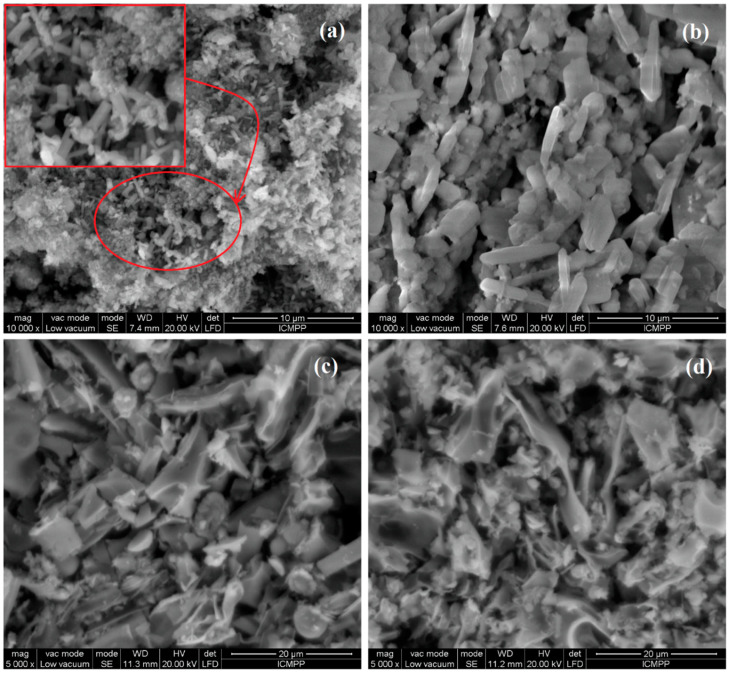
Scanning electron microscope (SEM) images of the ZnO (S1) (**a**), MoO_3_ (S2) (**b**), ZnO:MoO_3_ (S4) (**c**) and ZnO:MoO_3_ (S5) (**d**) nanostructures.

**Figure 2 nanomaterials-10-01476-f002:**
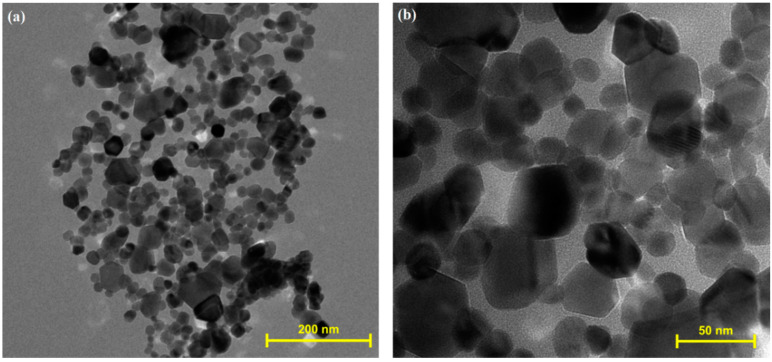
Transmission electron microscope (TEM) micrographs of nanocrystalline building blocks on sample (**a**) S5 × 350 k; (**b**) × 1300 k.

**Figure 3 nanomaterials-10-01476-f003:**
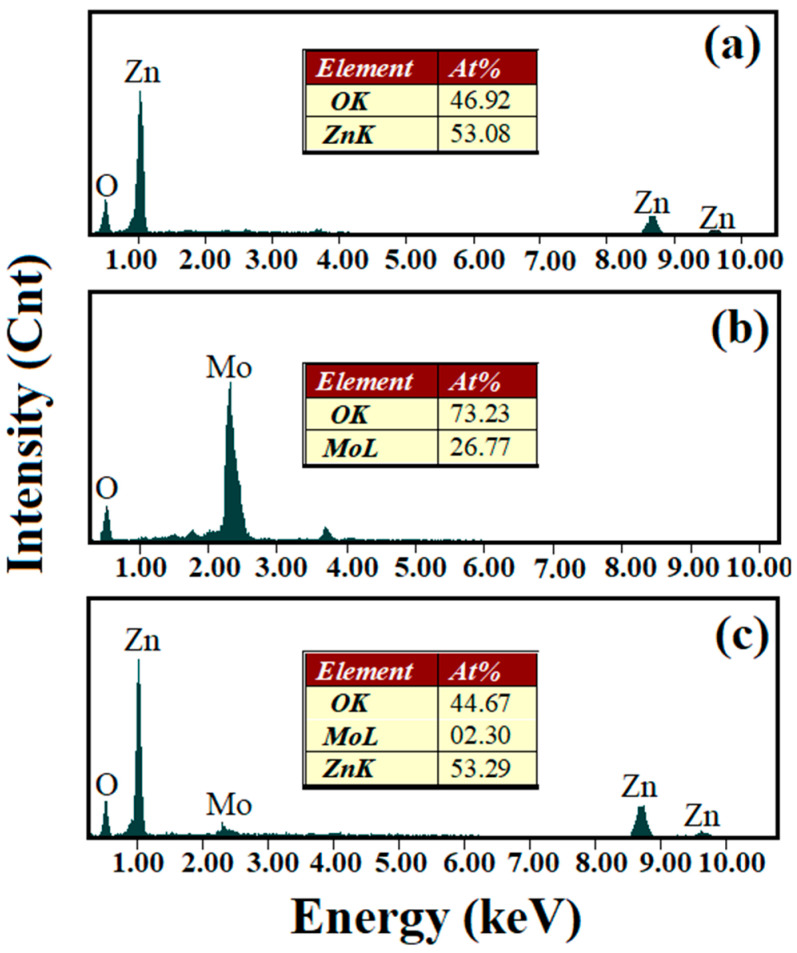
EDX spectra of pure ZnO (**a**), MoO3 (**b**) and ZnO:MoO3 (S5) (**c**) nanostructures.

**Figure 4 nanomaterials-10-01476-f004:**
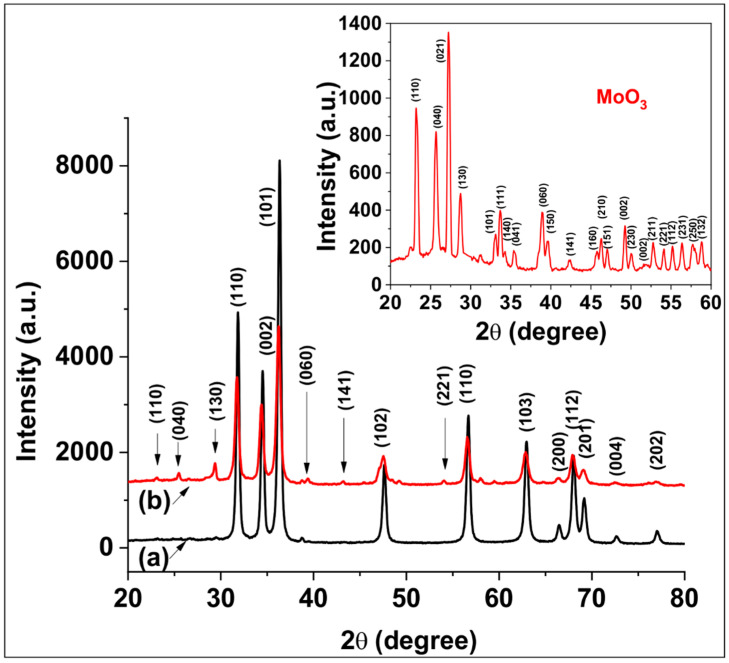
X-ray diffraction (XRD) patterns for pure ZnO (**a**), MoO_3_ (**inset**) and ZnO:MoO_3_ (S5) (**b**) composite material.

**Figure 5 nanomaterials-10-01476-f005:**
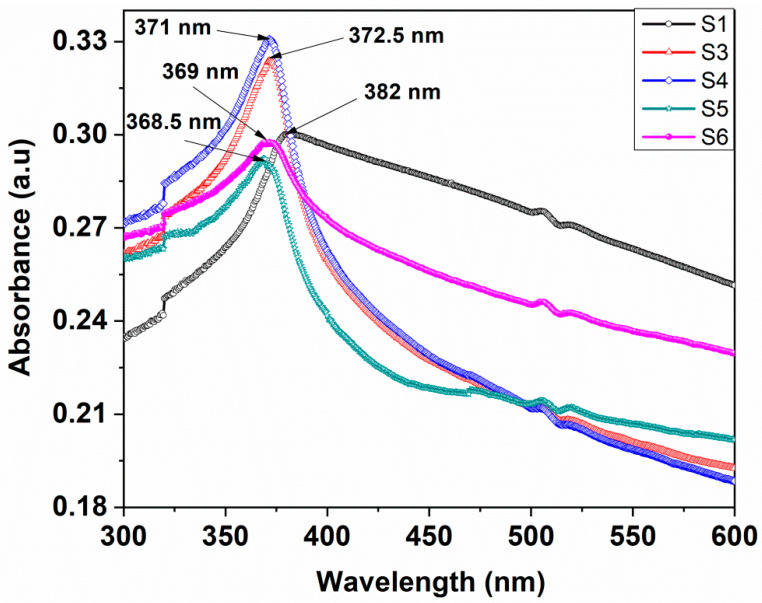
Absorption spectra of pure ZnO and ZnO-incorporated MoO_3_ nanostructures with different atomic weight percentages of MoO_3_ oxide.

**Figure 6 nanomaterials-10-01476-f006:**
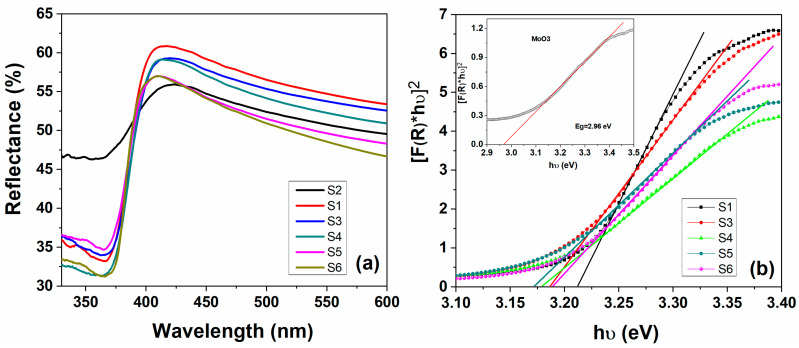
Diffuse reflectance ultraviolet–visible (UV–vis) spectra (**a**) and plots of ${\left[F\left(R\right)h\vartheta \right]}^{2}$ versus *hυ* (**b**) for pure ZnO (S1), MoO_3_ (S2) (**inset**), and ZnO:MoO_3_ (S3 → S6) nanostructures.

**Figure 7 nanomaterials-10-01476-f007:**
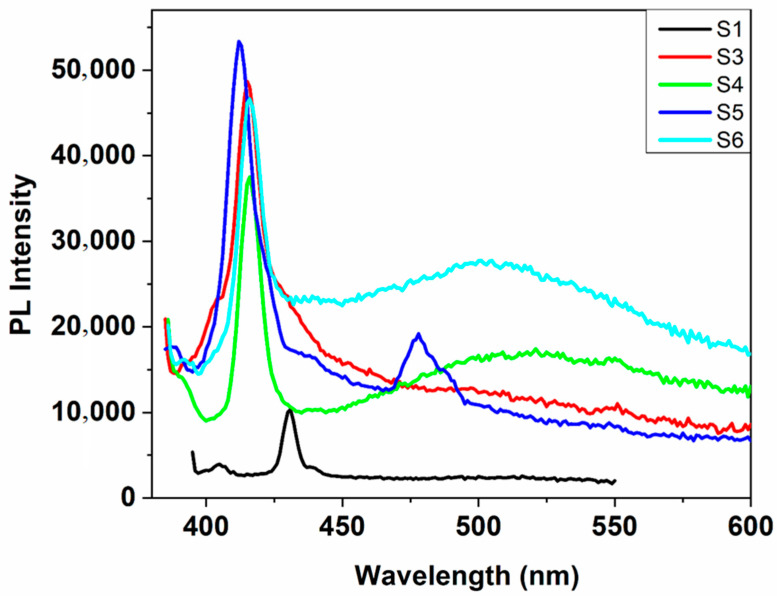
Fluorescence spectra of pure ZnO and ZnO:MoO_3_ nanostructures, recorded in 1-propanol solution, at room temperature.

**Figure 8 nanomaterials-10-01476-f008:**
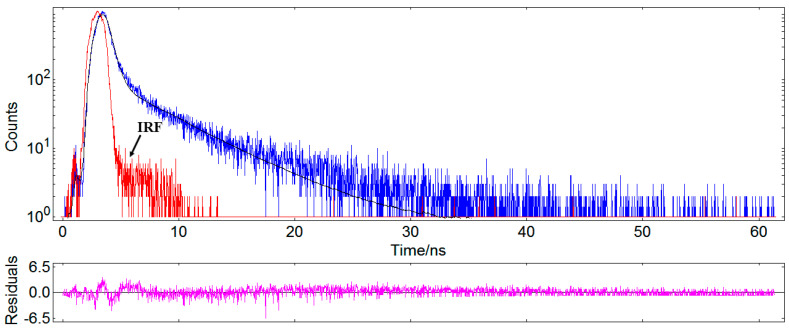
The lifetime decay profiles for the ZnO:MoO_3_ (S5) sample (λ_ex_ = 406 nm); IRF, instrument response function.

**Figure 9 nanomaterials-10-01476-f009:**
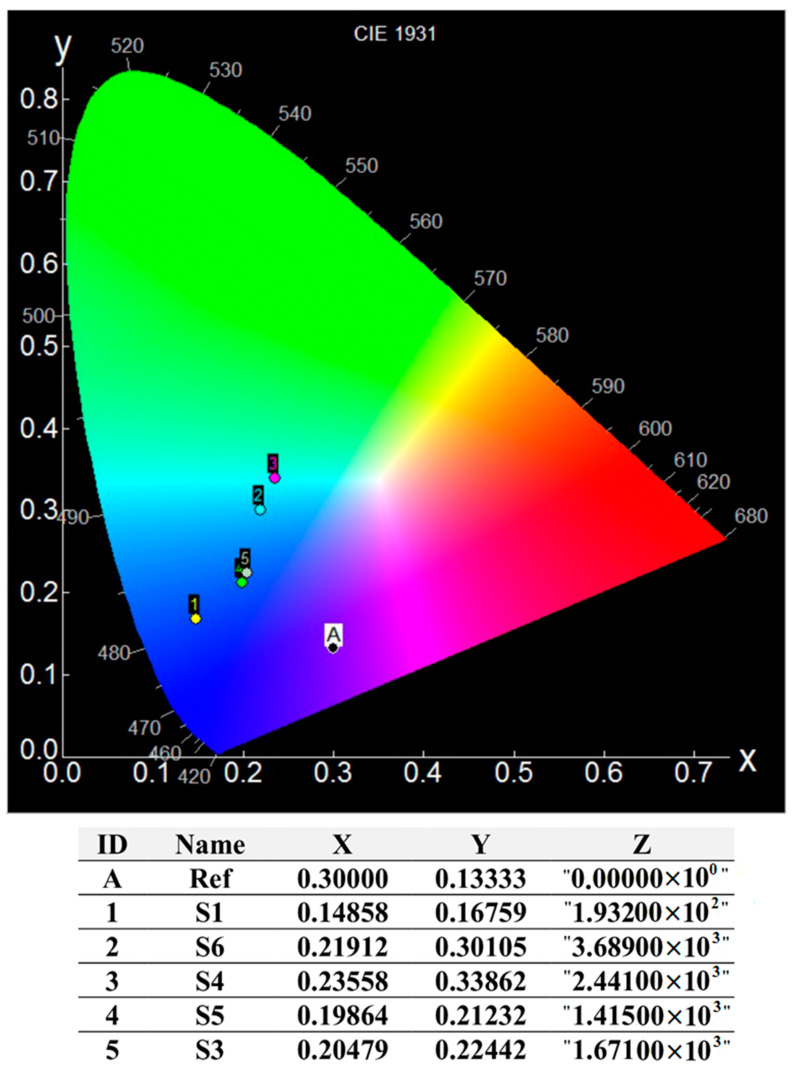
The CIE color chromaticity diagram of pure ZnO and MoO_3_-incorporated ZnO nanostructures.

**Figure 10 nanomaterials-10-01476-f010:**
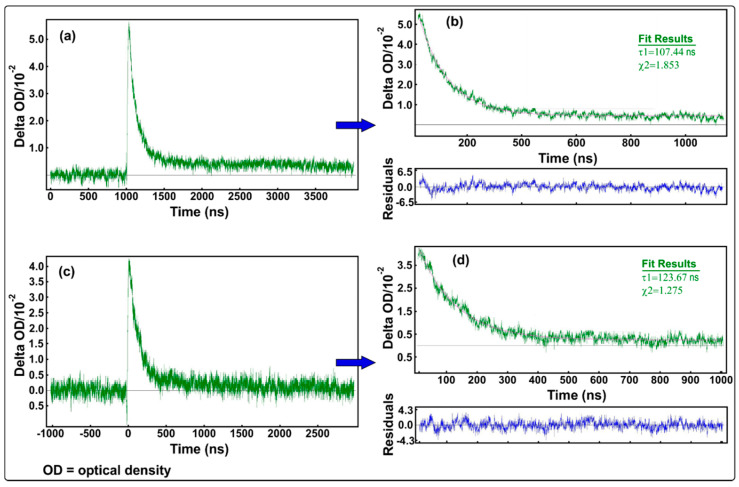
Kinetics transient absorption decay and exponential fitting for ZnO:MoO_3_ (S3) (**a**,**b**) and ZnO:MoO_3_ (S5) (**c**,**d**) samples, upon excitation at 372 nm.

**Figure 11 nanomaterials-10-01476-f011:**
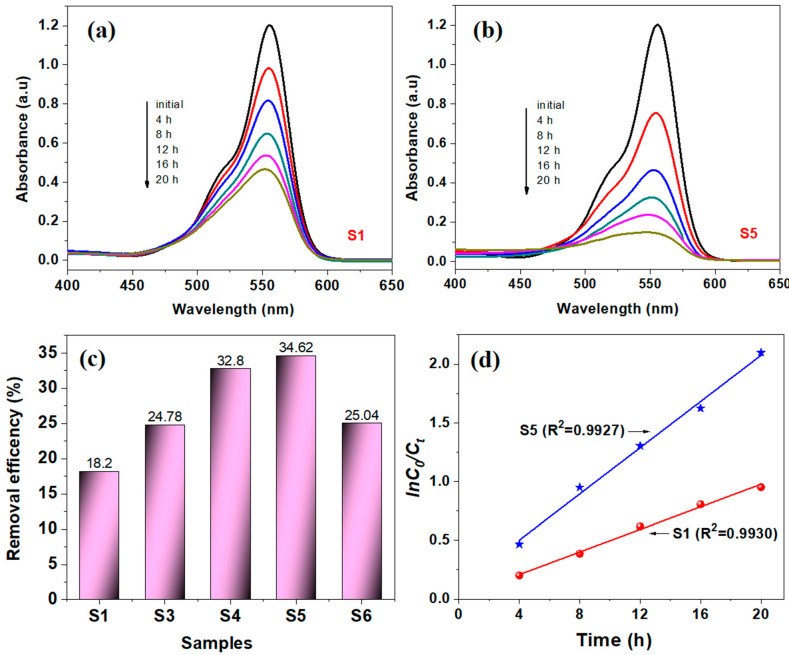
The absorption spectra changes of the RhB dye in aqueous solution, at various times, upon exposure to visible light irradiation in the presence of pure ZnO (S1) (**a**) and ZnO:MoO3 (S5) (**b**) samples; Removal efficiency (%) of the degradation of RhB dye, after 6 h of visible light irradiation (**c**); Pseudo-first order kinetics for RhB dye degradation in the presence of S1 and S5 samples, respectively (**d**).

**Table 1 nanomaterials-10-01476-t001:** Results of EDX (energy-dispersive X-ray spectroscopy) measurements for the pure ZnO, MoO_3_, and ZnO-incorporated MoO_3_ nanostructures.

Sample Codes	Composition of Precursor Salt	Atomic Molar Ratio Mo/Zn (%)	Atomic Concentration (%) Obtained from EDX		
			Zn (%)	O (%)	Mo (%)
S1	0.4 g Zn(NO_3_)_2_·6H_2_O	-	53.08	46.92	-
S2	0.4 g (NH_4_)_6_Mo_7_O_24_·4H_2_O	-	-	72.23	26.77
S3	0.4 g Zn(NO_3_)_2_·6H_2_O 0.002 g (NH_4_)_6_Mo_7_O_24_·4H_2_O	0.84	51.20	47.85	0.92
S4	0.4 g Zn(NO_3_)_2_·6H_2_O 0.004 g (NH_4_)_6_Mo_7_O_24_·4H_2_O	1.66	53.86	44.62	1.51
S5	0.4 g Zn(NO_3_)_2_·6H_2_O 0.008 g (NH_4_)_6_Mo_7_O_24_·4H_2_O	3.26	53.29	44.67	2.30
S6	0.4 g Zn(NO_3_)_2_·6H_2_O 0.012 g (NH_4_)_6_Mo_7_O_24_·4H_2_O	4.81	50.55	44.62	4.82

**Table 2 nanomaterials-10-01476-t002:** Calculated band gap energies and band edge potentials for pure ZnO and ZnO-incorporated MoO_3_ nanostructures.

Codes	Eg	E_VB_ (eV)	E_CB_ (eV)
**S1**	3.210	2.890	−0.310
**S3**	3.186	2.883	−0.303
**S4**	3.177	2.878	−0.298
**S5**	3.170	2.875	−0.295
**S6**	3.189	2.884	−0.304

**Table 3 nanomaterials-10-01476-t003:** Fitted decay times and average lifetimes ($\tau_{eff}^{*}\;\left(ns\right)$) for all pure ZnO and ZnO:MoO_3_ nanostructures.

Codes	Fast Decay Component		Slow Decay Component		χ^2^	$\tau_{eff}^{*}\;\left(ns\right)$
	τ_1_ (ns)	a_1_ (%)	τ_2_ (ns)	a_2_ (%)		
**S1**	0.0664	74.06	4.1058	25.94	1.004	3.93
**S3**	0.1068	51.95	5.3416	48.05	1.034	5.23
**S4**	0.0624	76.76	4.6088	23.24	0.995	4.42
**S5**	0.4654	72.68	5.0603	27.32	1.007	4.17
**S6**	0.0662	74.05	4.0695	25.95	1.086	3.89

**Table 4 nanomaterials-10-01476-t004:** Comparison of photocatalytic activities of ZnO:Mo materials.

Photocatalyst Type	Type and Concentration of Dyes	Amount Photocatalyst (g/mL)	Light Source	Reaction Rate Constant *k* (min^−1^)	Ref.
ZnO@h-MoO_3_	Methylene Blue	0.5	Vis	-	[14]
ZnO:Mo (2%)	Orange II	0.05/80	UV	0.0032	[27]
ZnO:Mo (0.6%)	Direct Yellow 27	1/1000	Vis (500 W)	0.0007	[36]
	Acid Blue 129	1/1000		0.00085	
MoO_3_/ZnO	Methylene Blue	-	UV	0.00138	[37]
ZnO:MoO_3_	Rhodamine B	0.005/10	Vis (100 W)	0.0018	This work

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
