# Peer review of "New Electrospun ZnO:MoO3 Nanostructures: Preparation, Characterization and Photocatalytic Performance"

_nanomaterials, 2020, doi:10.3390/nano10081476_

Round 1

Reviewer 1 Report

The manuscript deals with preparation (by electrospinning methods), characterization and photocatalytic performance of Zn and Mo mixed oxides nanostructures.

The authors conclude that the RhB degradation efficiency of ZnO is increased by MoO3 incorporation, with the best performance obtained with 2% of MoO3.

In my opinion the paper provides some useful information on the subject, but it is quite weak in some points. Therefore, revision is mandatory before publication.

Key points:

- SEM and XRD: on the basis of these data (Figs 1-3) the authors state the successful formation of the nanocomposite. Really this is not so direct, being somehow questionable. The differences in the crystallite sizes are really small and hardly correlated to the defectivity of the mixed oxides. Moreover, on which basis the formation of MoO3/ZnO heterojunctions was proved? This is a very important point to explain photocatalytic activity of these composites. In my opinion TEM data are then mandatory.

- Optical properties: the differences in the band gaps are small. It is well known that the extrapolation of the band gap value by the Kubelka-Munk function is affected by some error. What the error in your experiments? For instance, why you reported a band gap of 3.20 eV for ZnO whereas the extrapolated value from Fig. 5 is around 3.21 eV?

Another unclear point is the presence of the strange band at 478 nm (Fig. 6) only on the 2%Mo sample. Why? Why not in the 1% sample? Please consider that these two samples exhibited quite similar SEM photos and quite similar higher removal efficiency. This must be carefully discussed.

- Photoluminescence data and CIE diagrams are not well explained and there is a lack of discussion and correlation with photocatalytic data, appearing quite useless if not related to the observed photocatalytic performances.

- Photocatalytic experiments: which is the reproducibility of experiments and the calculated kinetic constants. What about the deactivation of the composites?

- Finally, it is important to compare data of these composites with other systems reported in the literature. I suggest to include a table with main results. The authors should also stress the real difference with other ZnO/MoO3 nanostructures prepared by different approaches than electrospinning.

Minor corrections:

- Introduction: the list of the improvements due to metal doping (lines 34-38) must be written in a more uniform and fluent way. You instead find: i) increasing …. (ii) decrease …. (iii) by introducing.

- Some general references on the use of MoO3 as photocatalyst must be provided (line 48).

- Line 57: “electrospinning” instead of “electrospinning”

- Page 15: Lines 281-283 are a repetition of lines 278-281. Please remove

Author Response

Please see the attachement,

Reviewer 2 Report

The manuscript «New electrospun ZnO:MoO3 nanostructures: Preparation, characterization and photocatalytic performance» by P. Pascariu, M. Homocianu, N. Olaru, A. Airinei and O. N. Ionescu is devoted to the preparation, characterization and photocatalytic Rhodamine B dye degradation efficiency of the ZnO:MoO3 materials of different compositions. This is a high-level work, which has great importance for the wastewater treatment. However, the manuscript needs major revision and cannot be accepted in its current form. The authors should correct the following points.

1. I recommend to change the labeling for the doped samples (without brackets), and then use it instead of the bulky and frequently "MoO3 Ë— incorporated ZnO nanostructures".
2. Move the Figures 6 and 7 above, after their first mention in the text.
3. Text has some incorrect, unclear sentences, as well as typos (some of them are marked by highlighter in the attached pdf-file).
4. Authors specify, that “the development of ZnO:MoO3 nanostructures by electrospinning-calcination method and then their testing for dye degradation have not been reported in literature so far”. However, authors missed very important references to site. (Materials Letters 262 (2020) 127049, J. Mater. Chem., 2011,21, 4217-4221; Chemical Engineering Journal 330 (2017) 322–336323) Some of them are devoted to the synthesis of ZnO:MoO3 structures and photo degradation of dyes.
5. Page 3, line 83. I advise to remove “sol. A”, since you never mentioned it again.
6. When you write “ZnO:MoO3 (0.5%)” Please, specify, which kind of percentage (mass, molar?) and the percentage of what is this?
7. Table 1. Add “MoO3" to description, it is better to compare the experimental measurements of atomic concentrations with calculated ones.
8. Page 6, lines 152-155. The corresponding percentage in Table 1 is given for Mo! Not for ZnO:MoO3 composite!
9. Figure 1. Please specify which kind of MoO3? (a,h?) Please change the composite description from "ZnO:Mo3" to “ZnO:MoO3”
10. Figure 2. I think it would be better to omit this figure.
11. Page 9, line 187. Why do you mention Ni2+ ionic radius?
12. Table 2. Please give the explanation to the increase of the optical band gap for ZnO:MoO3 (3%) compared to other composites.
13. Page 15, lines 278 and 284, please provide the decryption of “PL” and “I”.
14. Figure 7. I think it would be better to omit this figure. The fitting curve is not visible.
15. Lines 300 and 317-318 – there is some inconsistency.
16. Figure 8. Probably it is also better to omit the figure.
17. Ref. 15. Please add the page numbers ("5447-5451").

Please, read also my additional comments in the attached pdf-file.

Reviewer 3 Report

New ZnO:MoO3 nanostructures that can be useful as effective photocatalytic materials for energy and environmental applications were created and investigated in the work. To prepare these materials, a relatively simple method of electrospinning was used. The authors conducted extensive research on the optical and catalytic characteristics of new materials, which corresponds to the subject of a special issue. In the study of these properties, a correlation of photocatalytic activity with optical and local structural characterizations was established. However, the correlation of these properties with the nanostructural characteristics of new materials is not fully described.

Authors should clarify the following important issue. It is not clear from the article how the chemical state of Mo atoms was determined, which was shown as MoO3. The molybdenum atoms/ions may have different oxidation states and they may incorporate into and distort the lattice of the ZnO crystal, or form MoO3 nanoinclusions/nanophases/clusters. In this case, the authors do not report anything about the size of MoO3 nanocrystals, but characterize the structure as nanocomposite/hybrid, containing MoO3/ZnO heterojunctions. The annotation of the article reports that the material was studied by transmission electron microscopy. However, these results are absent in the article. Information on the effect of Mo concentration on the nanostructured characteristics of the materials created should be presented more clearly. It would be interesting to know what state of the material is formed immediately after synthesis by electrospinning method and what processes/driving forces cause the formation of a two-phase ZnO:MoO3 composite during calcination.

The article contains the repetitions of information and the inaccuracies that need to be corrected. For example:

Line 278. “Instead, in the PL spectrum of pure ZnO nanostructures, the wavelength of the maxima of the emission band is located at 431 nm, with low intensity, and the green emission band has disappeared (compared to the spectra of MoO3 -incorporated ZnO samples). Instead, the undoped zinc oxide nanostructure displays only an emission band around 431 nm, with low intensity and the green emission band disappeared in comparison with MoO3-incorporated ZnO samples.”

Fig. 1. ZnO:Mo3 (?)

Fig. 7. It is advisable to clarify what the IRF curve means, and check the sentence “an excitation wavelength of 406 nm”…

Line 371 -372:  MoO3 and  MoO3

Round 2

Reviewer 1 Report

The authors revised the manuscript accordingly to the suggestions of reviewers. I find that the quality of the manuscript has been improved by the revision made. I have only a request that the authors should address before publication. Namely, TEM analysis points to the presence of various particles with different sizes. In my opinion it is necessary to look at the diffraction patterns to discriminate between the two oxides and to verify the formation of heterojunctions. This would greatly increase the value of this paper.

Reviewer 2 Report

The authors adequately revised and have corrected the current version of the manuscript «New electrospun ZnO:MoO3 nanostructures: Preparation, characterization and photocatalytic performance» and correctly answered on the raised questions and remarks. Thus, the manuscript can be accepted for publication in Nanomaterials in its current form, but some polishing of the English is still required.

Reviewer 3 Report

The manuscript has been significantly improved after revision. However, the authors should describe how the samples for TEM studies were prepared. It is also important to discuss how the TEM and XRD results correlate. (TEM studies have shown that the ZnO: MoO3 material contains nanocrystallites with sizes of 20-30 nm and 40-45 nm. According to XRD studies, the average crystallite size is D = 14.93 nm.)  For the presented TEM results, it is not clear which crystallites correspond to ZnO and/or to MoO3 phases.  It is possible that a single crystallite with a size of more than 20 nm consists of a mixture of ZnO (the size is ~ 15 nm, according to XRD) and MoO3. So, for the article, the addition of SAED (selected area electron diffraction) patterns for some nanocrystallites is desirable.

The abstract needs to be corrected: “The obtained electrospun  ZnO:MoO3 hybrid materials were characterized by X-ray diffraction (XRD), scanning and transmission electron microscopies….”
